# A Review of Bioactive Compound Effects from Primary Legume Protein Sources in Human and Animal Health

Zachary Shea [1], Matheus Ogando do Granja [2], Elizabeth B. Fletcher [2], Yaojie Zheng [2], Patrick Bewick [2], Zhibo Wang [2,3], William M. Singer [4] and Bo Zhang [2,*]

1. United States Department of Agriculture–Agricultural Research Service, Raleigh Agricultural Research Station, Raleigh, NC 27606, USA; zachary.shea@usda.gov
2. School of Plant and Environmental Sciences, Virginia Tech, Blacksburg, VA 24061, USA; matheusogranja@vt.edu (M.O.d.G.); fbodie1@vt.edu (E.B.F.); yaojie@vt.edu (Y.Z.); pwb@vt.edu (P.B.); zhibowang@danforthcenter.org (Z.W.)
3. Donald Danforth Plant Science Center, Olivette, MO 63132, USA
4. Center for Advanced Innovation in Agriculture, Virginia Tech, Blacksburg, VA 24061, USA; wilmsing@vt.edu
* Correspondence: zhang76@vt.edu

**Abstract:** The global demand for sustainable and nutritious food sources has catalyzed interest in legumes, known for their rich repertoire of health-promoting compounds. This review delves into the diverse array of bioactive peptides, protein subunits, isoflavones, antinutritional factors, and saponins found in the primary legume protein sources—soybeans, peas, chickpeas, and mung beans. The current state of research on these compounds is critically evaluated, with an emphasis on the potential health benefits, ranging from antioxidant and anticancer properties to the management of chronic diseases such as diabetes and hypertension. The extensively studied soybean is highlighted and the relatively unexplored potential of other legumes is also included, pointing to a significant, underutilized resource for developing health-enhancing foods. The review advocates for future interdisciplinary research to further unravel the mechanisms of action of these bioactive compounds and to explore their synergistic effects. The ultimate goal is to leverage the full spectrum of benefits offered by legumes, not only to advance human health but also to contribute to the sustainability of food systems. By providing a comprehensive overview of the nutraceutical potential of legumes, this manuscript sets a foundation for future investigations aimed at optimizing the use of legumes in the global pursuit of health and nutritional security.

**Keywords:** bioactive compounds; legumes; peptides; protein; isoflavones; antinutritional factors; saponins

## 1. Introduction

The market for plant-based protein is rapidly expanding and is projected to reach about $42.5 billion by 2034, driven by increasing popularity in developed countries, including the United States and the United Kingdom [1]. The major crops that are used to produce plant-based protein include soybean (*Glycine max*), pea (*Pisum sativa*), chickpea (*Cicer arietinum*), and mung bean (*Vigna radiata*) [2–5]. Of these, soybean and pea are the most prominent. These plant proteins are used to develop products including tofu, meat analogues (e.g., plant-based burgers), milk substitutes, and protein powder [6,7].

Consumer demand for plant-based protein, specifically from legumes, has increased for a variety of reasons, including environmental sustainability, low cost of production, nutrition and health benefits, and ethical concerns in animal protein production [1,3]. Arguably, the primary reason is due to plant-based protein being more environmentally sustainable than animal-based protein. Animal-based protein tends to have a much higher impact on climate and the environment than plant-based protein due to higher water usage and greenhouse gas production [8–10]. As consumers' interests in environmentally friendly solutions grow, it can be expected for plant-based alternatives' demand to grow. The health

and nutritional benefits are an increasingly prominent secondary reason for the increased demand of plant protein. Diets high in plant proteins have been found to lower risk of cardiovascular diseases, obesity, diabetes, and other metabolic features [11].

Although present in small quantities, bioactive compounds in plant-based foods are crucial for health benefits [12]. Legumes, as significant sources of plant-based protein, are rich in bioactive compounds such as isoflavones, bioactive peptides, protein subunits, antinutritional factors, saponins, and galactosides [13,14]. Bioactive compounds exhibit a diverse array of health-promoting effects, with their specific benefits varying according to the type of compound [15]. These benefits consist of the following: reducing inflammation, triglycerides, and metabolic issues; exhibiting antidiabetic, anticancer, and antioxidative effects; enhancing cardiovascular, bone, and cognitive health through isoflavones; and anticancer properties [14,16,17].

The broad spectrum of potential health benefits attributed to these compounds has significantly sparked interest among researchers and scientists eager to harness their power to efficiently bolster human and animal health. The potential pharmaceutical uses of these compounds open up a broad field for research, demanding an intricate understanding of bioavailability, metabolic pathways, and interactions within human physiological frameworks. This is especially pertinent for lesser-studied legumes like mung beans and chickpeas, which may contain bioactive compounds with efficacy rivaling or surpassing those found in soybeans. With the expanding body of knowledge on these bioactive compounds and the growing reliance on plant-based proteins, their application in health and nutrition is anticipated to rise accordingly.

Although previous reviews have highlighted leguminous bioactive compounds, they predominantly focused on bioactive peptides and did not include other compounds, such as saponins, protein subunits, and antinutritional factors [18–22]. Furthermore, while some reviews mentioned other bioactive compounds, this was typically limited to stating their presence in legumes. A direct bioactive compound content or functionality comparison between different leguminous species was not provided, and similarities between species were not discussed [23,24]. The only review that included a comparison between species focused on vegetables and did not discuss soybean, chickpea, and pea [25].

Therefore, this review aims to provide an in-depth exploration of the primary bioactive compounds in major legume-based protein sources (mainly soybean and pea as well as chickpea and mungbean) and compare the differences in bioactive compounds amongst these crops. Additionally, it will explore the function and importance of these compounds as well as the research conducted on animal and human health effects. Such endeavors are not only essential for advancing our grasp of plant-based nutrition but are also pivotal for leveraging the comprehensive benefits of legumes in developing functional foods and innovative therapeutic agents. By providing an in-depth examination of the primary bioactive compounds in major legume-based protein sources and comparing their bioactive profiles, this review sets a foundation for future research on their effects on animal and human health, contributing to the optimization of legumes in the quest for health and nutritional security.

## 2. Bioactive Peptides

### 2.1. Bioactive Peptides Importance and Overview

Bioactive peptides (BPs) are short chains of amino acids known for their physiological regulatory roles within the human body. Endogenously produced BPs are produced within the body, while exogenous BPs are obtained from food or dietary supplements and medications [26]. Most research has been centered around BPs that are derived from animal sources, but plants, specifically legumes, have been shown to be a good source for many of these molecules with beneficial effects on human health [27]. Compared to other plant species, legumes have a much higher amount of BPs and thus the focus of research on BPs in plants tends to be centered on legumes [28]. This review emphasizes plant-derived exogenous BPs from major legume sources. Most of the research in this area has focused

on soybeans, but insight is provided into BPs derived from mung bean, chickpea, pea, and cowpea.

BPs are usually encrypted within original proteins and do not become active until after digestion or fermentation [29]. BPs are more bioavailable and can be less allergenic than proteins due to their smaller size compared to primary proteins, and have a range of functions including antihypertensive, antioxidant, antimicrobial, anti-diabetic, and hypocholesterolemic activities, depending on the properties of the amino acids within the BP [18,27,30]. These BPs can have antihypertensive properties by lowering blood pressure through binding and inhibiting angiotensin I converting enzyme (ACE) [31]. BPs can have antioxidant activities due to some polar amino acids, histidine, such as lysine, glutamate, proline, aspartate, and tyrosine, being able to form hydrogen bonding at their C-terminus [32]. This allows them to serve as antioxidants because the polar residues enables the use of a hydrogen atom as a donor to scavenge for free radicals [32]. While not fully understood, it is believed that some BPs have antimicrobial properties due to the presence of positively charged amino acids, lysine and arginine. The positive charge allows them to bind to and cause damage to the negatively charged membrane of bacterial cells [28]. BP immunomodulatory and anticancer properties are still being determined, but as of now, it is largely deemed to be due to their specific secondary structure [28]. Consequently, these molecules find applications for both dietary and medicinal purposes.

The mass production of BPs for these markets mainly involves enzymatic hydrolysis using various enzymes in vitro (e.g., pepsin and trypsin), using microbial fermentation, or both [27,33,34]. Enzymatic hydrolysis can produce BPs by using various proteolytic enzymes to break proteins down into BPs [35]. While this process can produce BPs that are safe for consumption, it unfortunately has very low yield, and enzymes are expensive [36]. Microbial fermentation provides a cheap way to produce BPs by using lactic acid bacteria to break down proteins into peptides [36]. Unfortunately, this method also has low yield, and it is difficult to produce specific BPs of interest [37]. This, in turn, has prompted further research to look into alternative methods of production. Specific BPs can be chemically synthesized using recombinant DNA technology or post-purification engineering [38,39]. The desired end-product determines which method and specific conditions are used, and their activity and functionality is determined by their amino acid sequence and composition [27,29].

## 2.2. Comparison of BPs across Species

Each BP may serve several different functions. Many BPs with a wide array of functions have been identified from soybeans, but other legume crops have shown promise as sources of these plant-based therapeutic compounds. While there is not as much research available on BPs from other species, it is possible that they contain BPs with similar properties. In this review, several functions of BPs are compared, and examples found in soybean, chickpea, pea, mung bean, and cowpea are provided. Table 1 presents a list of these identified BPs.

**Table 1.** Functionality and source of various bioactive peptides found in plant proteins.

| Activity | Source | Peptide Name | Reference |
|---|---|---|---|
| Antihypertensive | Soybean | IY and WMY | [40] |
| Antihypertensive | Mung bean | LRLESF, HLNVVHEN, PGSGCAGTDL, and LPRL | [41] |
| Antimicrobial | Pea | BCBS-11, LSDRFS and SDRFSY | [42,43] |
| Antimicrobial | Chickpea | Leg1 and Leg2 | [44] |
| Antimicrobial | Soybean | NuriPrep 1653 | [45] |
| Antioxidative | Soybean | FDPAL | [46] |
| Anticancer | Soybean | Lunasin | [47] |
| Hypocholesterolemic | Cowpea | GCLTN | [48] |

**Table 1.** *Cont.*

| Activity | Source | Peptide Name | Reference |
|---|---|---|---|
| Hypocholesterolemic | Soybean | VAWWMY | [49] |
| Immunomodulatory | Pea | Psd1 | [50] |
| Immunomodulatory | Soybean | Soymetide-13 | [51] |
| Antioxidative | Chickpea | NFYHE, ALEPDHR, LTEIIP, RQSHFANAQP | [52,53] |
| Anticancer | Chickpea | ARQSHFANAQP | [54] |
| Antidiabetic | Chickpea | GKGSGAF, RASAAGGGGGGVSSR, QNPLSSAAPTGAGKPY, AMMELGWSTSGEFLL | [55] |
| Antihypertensive | Chickpea | MDL, MDLA, MD, and MDPLI | [56] |
| Antioxidative | Pea | YSSPIHIW, ADLYNPR, HYDSEAILF, AGVLPGIK and GHYPNPDIEYG | [57] |
| Antihypertensive | Pea | LGP, LKP, YW, VY, AKSLSDRFSY, LSDRFS, SDRFSY | [58] |
| Antidiabetic | Pea | ALP, LLP, VLP, and SP | [59] |

Many plant-derived BPs are capable of interfering with the growth and prevalence of microorganisms. These molecules make up the endogenous plant defense mechanisms which are used to fight against pathogenic action. Typically ranging from 10 to 50 amino acids long, these peptides possess positively charged amino acids, allowing for the penetration of membranes and homeostasis disruption [60–62]. BCBS-11, a peptide derived from soybean, has been shown to disrupt biofilm and bacterial membranes, suggesting a potential dental application [42]. These molecules can also work as a food preservative [63]. Chickpea peptides Leg1 and Leg2 display antibacterial and antifungal activities [44]. A peptide from pea protein called NuriPrep 1653 has proven to be effective in controlling the multidrug-resistant pathogen *Acinetobacter baumannii* [45]. Additionally, the peptides LSDRFS and SDRFSY from pea were found to reduce infection from SARS-CoV-2 [43]. A major benefit of using antimicrobial BPs over traditional antibiotics is that they have been shown to have a decreased chance of creating antibiotic resistance in pathogens [63].

Chronic high blood pressure can lead to a number of health issues including cardiovascular disease, stroke, and arteriosclerosis. One proposed mechanism of regulating blood pressure is the inhibition of ACE, which is known to raise blood pressure through its activity. Along with many other BPs identified in soybean, IY and WMY extracted from soybean protein show strong ACE-inhibition activity [40]. LRLESF, HLNVVHEN, PGSGCAGTDL, and LPRL BPs extracted from the <1 kiloDalton (kDa) peptide fraction of hydrolyzed mung bean meal were shown to have a similar effect [41]. The peptides, MDL, MDLA, MD, and MDPLI from chickpea, and LGP, LKP, YW, VY, AKSLSDRFSY, LSDRFS, and SDRFSY from pea, have been shown to significantly decrease ACE activity in vitro, but have yet to be used in any animal or human studies [56,58]. Plant proteins are good sources of these BPs because they have less cholesterol and fat than animal foods. While synthetic ACE inhibitors have been reported to cause several uncomfortable side effects, plant-derived alternatives such as BPs have been shown to be effective with few side effects [64,65]. These peptides must be relatively short (2–12 amino acids), or else they are unable to bind with the active sites of ACE [18]. This ability is also strongly associated with the occurrence of aromatic, hydrophobic, and basic amino acids at the C-terminal [66,67].

Numerous plant-derived BPs have been found to reduce levels of lipids and cholesterol in the blood by interacting with bile and cholesterol in the gut and mediating the action of hormones, receptors, and genes related to the processing and expulsion of cholesterol and lipids [68–70]. These BPs usually contain hydrophobic amino acids that allow the molecule to be amphipathic, so it can disrupt micellar cholesterol and increase its solubilization [71]. One such BP is GCTLN, which is found in cowpea. In addition, some BPs can bind to bile acids in the system and prevent their absorption into the blood stream, leading to reduced plasma cholesterol levels [48,72,73]. VAWWMY from soybean showed a bile-binding ability nearly as strong as a hypocholesterolemia medicine [49].

An abundance of free radicals in the body can lead to oxidative stress, leading to diseases such as Parkinson's, cancers, diabetes, and atherosclerosis [18]. Oxidative stress can also cause cellular damage that accelerates aging [74]. Aromatic amino acids like tyrosine and phenylalanine allow BPs to scavenge free radicals through proton donation, while cysteine, tryptophan, and histidine act through electron transfer [75,76]. FDPAL, a soybean-derived BP, has been shown to be capable of scavenging free radicals in vitro [46]. These molecules can also prevent lipid oxidation and reduce microbial growth, making them useful as a food preservative [77]. In pea, the peptides YSSPIHIW, ADLYNPR, HYDSEAILF, AGVLPGIK, and GHYPNPDIEYG were found to have high free radical scavenging rates in silico, while multiple peptides from chickpea were found to have free radical scavenging activity in vitro [53,54,57]. While the exact peptide sequences were not determined, mung bean was found to have some BPs that can serve as antioxidants by stabilizing some free radicals in vitro [78]. Some peptides in this class, like the soybean BP lunasin, are even capable of defending against some types of cancers [47]. For instance, peptides from chickpea have been found to inhibit the proliferation of human breast cancer cells by increasing the amount of p53, a protein that helps prevent tumor formation [54].

Responses to pathogens in the body are generated by an interconnected web of organs, tissues, and cells comprising the immune system. Immunomodulatory BPs interact with the immune system and can improve its ability to fight off infection by supporting adaptive immune responses through the targeting and upregulating of agents within the body, such as natural killer cells, lymphocytes, and macrophages. The defensin peptide Psd1 in pea has been shown to activate immune cells and increase the immune response of human immune cells [50]. Soymetide-13, a BP from soybean, has been shown to stimulate phagocytosis, which allows the body's immune cells to identify and intercept potential pathogens in the body [51]. Most BPs in this category contain hydrophobic, aromatic, and negatively charged amino acids, which allow them to interact with signaling pathways within the immune system [18].

Some BPs also possess antidiabetic properties. This mechanism is not fully understood but it is believed to be from the BPs' promotion of insulin release and glycogen synthesis [59]. This was identified in pea BPs that were fed to diabetic mice [59]. BPs from chickpea are believed to have antidiabetic effects by inhibiting enzymes related to type 2 diabetes [55].

*2.3. Uses of BPs in Research and Future Directions*

As previously discussed, BPs exhibit a wide range of uses and, unsurprisingly, have been tested to determine their impact on a variety of aspects of human and animal health. A majority of the research has involved BPs from soybeans and their effects in mouse and rat studies. Generally, soybean BPs have consistently demonstrated the ability to lower blood pressure, triglycerides, and cholesterol levels, as well as possess anticancer, antioxidant, and anti-inflammatory effects [14,79]. In addition, soy BPs have been found to improve cognitive function in rats by preventing long-term memory loss and increasing neuronal survival likelihood [80]. Besides improving long-term memory, soy BPs were found to have preventative effects on some age-related cognitive diseases, such as Alzheimer's [81]. Other specific cases of the beneficial effects of soy BPs on humans include peptides from black seed coat soybeans having cytotoxic effects against human liver, lung, and cervical cancers and anti-inflammatory effects on post-menopausal women [82–84].

Another emerging area of interest regarding BPs is the effect on the gut microbiome. Some studies have found that soy BPs were able to significantly influence the type of microbiota that were found in mice, rats, and obese men [85–87]. In mice, it was found to decrease some harmful microbes while increasing some beneficial microbes that could help in reducing negative health concerns like obesity [86]. While soy BPs have been found to have an impact on the gut microbiome in some species, it is important to note that this is not unanimously reported. For instance, one study found that soy BPs did not have any effect on human infants' gut microbiota [88]. Pea was also found to have varying effects on the gut microbiome of humans; some studies found no effect on gut microbiota where

others found a significant impact in terms of increasing gut microbial diversity [89–91]. Lastly, chickpea peptides were found to have antioxidant effects and enhance the growth of some beneficial microbes for fecal fermentation, such as *Bifidobacterium*, *Veillonella*, and *Pediococcus* [92].

While it is known that other plants contain BPs with beneficial effects, there is little research looking at them being directly used in human and other animal studies when compared to soybean. Pea, chickpea, and mung bean have all been found to have peptides with anti-inflammatory, anticancer, and antioxidant activities and reducing hypertension, cholesterol levels, and some chronic diseases [2,54,78,93–95]. After soy, pea has been used the most for human studies. For instance, pea protein was found to reduce glycemia and stimulate the release of insulin in human adults in one study [96]. Given that many previously mentioned studies have found positive health effects from pea, chickpea, and mung bean BPs in vitro, it is projected that these BPs can also have these beneficial effects in humans and animals. That being said, there will have to be more research to elucidate the effects. Besides direct use, BPs from legumes could also be used as a framework for the production of synthetic BPs. One study was able to chemically synthesize a BP similar to the chickpea equivalent that still exhibited the same antioxidant and anticancer properties [54]. This is particularly interesting as current BP production through enzymatic hydrolysis and microbial fermentation has several major limitations, including low yield and BP specificity. Being able to chemically synthesize the specific BP of interest would at least get rid of one of the previously mentioned limitations. Since this process is still very new, more work needs to be performed before this is conducted on a larger scale.

Given the wide range of beneficial effects of these BPs, it is expected that their uses in research for human and animal health will increase. Given the implication that some of these biopeptides have anticancer activities, they could even be used in some cancer, hypertension, and diabetic co-treatments. If health products were to be developed from these BPs, soy BPs are likely to be prioritized as more research has identified their effects compared to those from pea, mung bean, and chickpea.

## 3. Protein Subunits

### 3.1. Protein Subunits Importance and Overview

Soy proteins can be categorized based on their solubility into two main groups, namely water-soluble albumins and salt-soluble globulins, with the latter comprising the majority and primarily functioning as storage proteins [97]. A more detailed classification is achieved through ultracentrifugation based on sedimentation coefficients, which has led to the identification of fractions such as 15S, 11S, 7S, and 2S [98]. The 2S fraction, containing the lightest molecules and about 20% of the total proteins, includes 2S globulins, cytochrome C, Kunitz trypsin inhibitor (KTI), and Bowman-Birk trypsin inhibitor (BBI) [99]. The 7S fraction, accounting for approximately 40% of the total protein, consists of α-amylase, β-conglycinin, lipoxygenase, and soy lectins. The 11S fraction, comprising 30% of the soy proteins, includes glycinin. The 15S fraction, primarily glycinin dimers, represents about 10% of the total protein [99,100].

Glycinin and β-conglycinin are the primary storage proteins in soybeans, classified into 11S and 7S classes, respectively. β-conglycinin is a trimeric glycoprotein composed of α, α', and β subunits with molecular weights of 67 kDa, 71 kDa, and 50 kDa, respectively [101]. The structure of β-conglycinin, elucidated through X-ray crystallography, was not clarified until the early 2000s. This technique isolated the β3 trimer from a mutant line, facilitating the crystallization of β-conglycinin [102]. Further studies revealed the structure of β-conglycinin as comprising three homotrimers and seven heterotrimers formed by the subunits [103]. Similarly, the structure of glycinin was resolved in 2001 [104]. Glycinin consists of five subunits, named $A_1aB_1b$, $A_1bB_2$, $A_2B_{1a}$, $A_3B_4$, and $A_5A_4B_3$, interlinked by disulfide bonds. Pairs of acidic and basic peptides (or acidic–acidic–basic peptides as seen in $A_5A_4B_3$) are linked by hydrophobic and/or hydrogen bonds. Therefore, three subunit pairs could form a trimer, and two trimers stack to form a hexamer [104].

The functional properties of soy protein play an important role in the food industry, by impacting production, processing, storage, and transportation. Glycinin and β-conglycinin, the predominant storage proteins, along with their subunits, are essential for soy protein functionality, affecting gel formation, emulsification, and foaming. Gelation is an important process for producing foods such as tofu, determined by the thermal behavior of these protein subunits. Upon heating, glycinin and β-conglycinin undergo unfolding, denaturing, and aggregation, leading to gel formation. However, their thermal properties differ. The denaturation temperature of glycinin is higher (75.7 °C) compared to β-conglycinin (60.5 °C) at a 1% weight concentration. Conversely, β-conglycinin exhibits slower rates of aggregation and densification [105]. Thermal aggregation kinetics reveal that β-conglycinin has a slower rate of aggregation and densification, with monomers still observed at 100 °C, which eventually form soluble aggregates of limited size. In contrast, the densification and nucleation processes of glycinin are quicker, resulting in insoluble aggregates with densified cores [105]. Thus, glycinin contributes to gel hardness, while β-conglycinin contributes to gel elasticity. Within glycinin, increased $A_3$ content enhances hardness, while $A_5A_4B_3$ dictates gel formation energy requirements, and A4 influences gel softness [106,107].

Soy protein is also important for the health of animals and humans. Intake of soy protein can reduce total cholesterol, low-density lipoprotein (LDL), and triglycerides, potentially lowering the risk of cardiovascular diseases and hyperlipidemia [108,109]. The roles of some subunits and peptides have been demonstrated. Hydrophobic peptides, rich in $A_{1a}$ and $A_2$ subunits of glycinin, can bind with bile acid to promote its excretion, which leads to the decrease in cholesterol synthesis in liver [110]. A later study further demonstrated that the α′ subunit of β-conglycinin might play roles in activating the LDL receptor increased expression to decrease LDL level [111]. The over accumulation or synthesis of cholesterol will lead to cholesteryl ester deposition in the arterial walls, causing cardiovascular diseases [112].

### 3.2. Comparison of Protein Subunits across Species

Soybeans and other legumes, such as peas, mung beans, cowpeas, and chickpeas, all belong to the Fabaceae family. Despite this, they vary significantly in their protein content and composition. Soybeans, for example, contain approximately 40% crude protein by dry weight, which is higher than the protein content in lentils (21–31%), mung beans (15–32%), and peas (24–30%) as reported by Shrestha et al. [101,113]. Legume proteins are categorized into subunits based on their sedimentation coefficients, with the 7S subunits known as vicilins and the 10.5S to 13.0S subunits referred to as legumins. Unlike soy proteins, lentils have a higher proportion of legumins compared to vicilins, whereas mung bean proteins predominantly consist of vicilins [114,115]. Peas exhibit a variable legumin to vicilin ratio, ranging from 0.4 to 2.0, and also contain a third subunit, convicilin, which has a molecular mass of about 70 kDa and can form trimers with other convicilins or heteromeric trimers with vicilins [116].

The subunit composition also differs among lentils, mung beans, and peas. For instance, lentil legumin includes three acidic subunits (47 kDa, 42 kDa, and 32 kDa) and two basic subunits (20 kDa and 18 kDa), whereas mung bean legumin comprises one 40 kDa acidic subunit and one 24 kDa basic subunit [114,115]. Lentil vicilin forms trimers composed of either 50 kDa monomers or 70 kDa convicilins, and mung bean vicilin consists of three 8S subunits (α, α′, and β) and a 7S subunit [113,117]. Pea vicilin, similarly, is a trimer made up of three subunits (α, β, and γ) with molecular weights of approximately 48–50 kDa [116]. The variance in protein content, composition, and subunits across different legume species not only affects their nutritional value but also offers a broad spectrum of applications in food production and processing. It was shown that the lentil protein isolate has a lower gelation concentration but weaker gel strength; a similar situation was also observed in peas, where increased convicilin content led to a higher gelation concentration [116,117]. The application of different ratios of legumin to vicilin is further determined by their different compositions.

### 3.3. Uses of Protein Subunits in Research and Future Prospects

Soy protein has been a dietary staple food source for centuries, with ongoing research aimed at its applications, processing improvements, and nutritional optimization through technological advancements. The molecular structure and physicochemical properties of soy protein subunits significantly influence their functionality in various applications. To date, the 11S has only been crystallized as a homohexamer containing the subunit A3B4 [104]. Given that the functional characteristics of soy protein are closely tied to its molecular composition and arrangement, a lack of understanding of the 11S protein can significantly affect its application in food processing, such as gelation and emulsification. Future research should thus continue to focus on exploring the molecular structure of soy protein.

Soy protein content can vary by up to 22% across different soybean varieties, highlighting the potential for breeding solutions [118]. Despite the achievements in breeding over the years, soy protein content remains a complex quantitative trait, which is negatively correlated with oil content and yield. Extensive research has been conducted on the genetic map of soy protein, and the genetic control over its subunits has been relatively well elucidated. The Soybase database (https://www.soybase.org/, accessed on 14 February 2024) has documented over 240 quantitative trait loci (QTLs) associated with protein content [119]. However, most genetic mapping has used biparental populations, with limited studies on whole-genome mapping, which restricts the exploration of rare alleles and haplotypes. The key to breaking the negative correlation between protein content and other traits requires leveraging rapid and cost-effective sequencing technologies for high-resolution, genome-wide mapping, and maximizing the use of germplasm resources to expand the diversity of mapping populations. Furthermore, elucidating gene function through techniques such as gene editing and clarifying the regulatory networks of protein content are also steps in fully understanding this trait.

Currently, 15 genes controlling the 7S subunit (CG-1 to 15) and 7 genes controlling the 11S subunit (Gy1 to 7) have been identified [120,121]. Manipulating these genes, while maintaining the globulin content constant, holds promise for adjusting the ratio of subunits according to specific applications.

Beyond breeding solutions, some research has been performed to elucidate the health impacts of these protein subunits. Vicilin in mung bean has been found to have multiple positive health impacts, including antioxidative, antihypertension, and some anti-proliferative effects against human cancer cells in vitro [122]. Mung bean vicilin was reported to have hypocholesterolemic activity through inhibiting the enzymes responsible for cholesterol synthesis in vitro [123]. Pea vicilin had some similar properties as mung bean vicilin, including antihypertensive properties by inhibiting ACE in vitro [124]. Additionally, pea vicilin was found to be involved in lipid and fat metabolism [125]. Lastly, protein subunits from chickpea, including vicilin and glutelin, were examined in piglet and rat feeding trials to determine negative health impacts. Glutelin was found to be the easiest to digest for rats, while vicilin was found to cause some minor immune response in piglets [126,127].

While protein subunits from legume have been found to have some beneficial health effects, almost all of the studies carried out have been in vitro. This highlights an important need for research to be conducted using these protein subunits in human and animal health trials to determine if they have any applied effects. This need becomes even more significant given that some of the subunits, such as vicilin, have already been found to trigger an immune response in animals. While the research does look promising, more work needs to be performed to ensure that these subunits do not cause any negative health effects for humans and animals. Furthermore, if these subunits can cause negative effects in animals such as piglets, then research will need to identify the species-specific effects.

## 4. Isoflavones

### 4.1. Isoflavones Importance and Overview

Isoflavones are a subcategory of flavonoids, a part of a larger compound group commonly known as phytoestrogens [128]. Phytoestrogens are a non-steroidal, naturally occurring compound similar to the estrogen hormone in vertebrates, and isoflavones are the most estrogenic of these flavonoids [128]. In plants, however, they are classified as phytoalexins and phytoanticipins, playing a vital role in the overall health and function of the plants [129]. They have been shown to be beneficial in plant defenses against viral, bacterial and fungal pathogens and a deterrent against herbivore consumption [129]. They are also important as a chemoattractant for beneficial insects and soil bacteria necessary for nodulation [130]. Isoflavones have also been found to be involved with root/shoot growth and root nodulation as the knockdown of a gene related to the biosynthesis pathway of isoflavones caused a shortening of the root and shoot growth and a decrease in root nodulation [131]. They have also been shown to have beneficial effects on humans and their microbiome [132].

Legumes are highly valued for their protein and oil seed content. While isoflavones have been shown to have a strong positive correlation to plant health and root growth, the same cannot be said for seed nutrient content. Several studies have found that the total isoflavone content of soybean seed has a negative correlation with seed protein and oil content [133–135]. However, one study conducted identified several soybean varieties with strong positive correlations between protein and total isoflavone content [133]. While this finding has yet to corroborated with additional studies, others have identified soybean varieties that contain high levels of isoflavones while maintaining moderate protein levels [133]. Unfortunately, no studies have examined the relationship between protein and oil content with isoflavone content in peas, chickpeas, and other legumes.

While the genotype, environmental factors, and genotype by environmental interactions play a large role in the concentration of isoflavone and yield individually, one study reported that they found no significant correlation between isoflavone content and yield [134]. The findings of said study, however, are contradicted by another that concluded that as yield increased, so did isoflavone content [136]. The contradiction of findings for both protein and yield in relation to isoflavone content serves to highlight the need for additional research of this bioactive compound in soybean and other legumes.

### 4.2. Comparison of Isoflavones across Species

Isoflavones are present in many Fabaceae plants, including bean, lentils, chickpea, clover, peanut and soybean [137]. While it is important to note that the type of processing method can impact isoflavone levels, soybeans generally have the highest concentration of isoflavones among legumes and are the most common dietary source for humans and animals [138,139]. The isoflavone content of beans, chickpeas, and lentils were found to vary significantly between the three legume species and also among the studied varieties of each group [140]. The majority of isoflavone research has primarily focused on the most highly consumed legume, soybean. Therefore, the current understanding of isoflavone content among the other 20 legumes consumed by humans is limited, despite there being established protocols for the quantification of isoflavone content [141,142]

The isoflavones with the highest concentrations in soybean are daidzein (DAI) and genistein (GEN). GEN is a molecule important for the biosynthesis of a variety of antimicrobial compounds [143]. Soybean seeds contain the highest concentration of GEN among the legumes, while chickpeas and mung bean contain negligible isoflavone contents [144,145]. Pea GEN levels vary greatly but typically fall somewhere in between the GEN levels of soybean and chickpea [146]. GEN has been shown to have a beneficial impact on human health, most often used in the treatment of cancers [147]. DAI, on the other hand, is the secondary metabolite of GEN, the most important for plant defense, and the form of phytoestrogen used in treatment of heart disease and menopausal symptoms.

Isoflavone concentration, including GEN and DIA, depends on growth stage, plant part, and growing conditions. In red clover (*Trifolium pratense*), the concentrations of isoflavones increased and decreased with changes in maturity [148]. Isoflavones are present in all plant tissue but in varying levels of concentration. During vegetative stages, the leaves and stems have the greatest concentration of isoflavones [148]. Once flowering begins, concentrations rapidly decreased through plant tissues but the leaves will continue to have the highest concentration compared to other tissues [148].

*4.3. Uses of Isoflavones in Research and Future Directions*

As phytoestrogens, isoflavones exhibit diverse effects on both animals and humans. In many species, the incorporation of or increase in isoflavones in the diet leads to both positive and negative effects on their health. Both the positive and negative outcomes of isoflavone consumption have been observed in humans, ruminant animals, and poultry.

In human studies, women often see more benefits of increased isoflavone content in their diet. When DAI was incorporated into the diets of menopausal women, it often reduced the severity of their symptoms. These improvements include, but are not limited to, hot flashes, mind fog, mood swings, and increased spinal bone density in post-menopausal women [149–153]. While the mechanism by which isoflavone increases bone density is not fully understood, multiple studies have linked increased isoflavone consumption with an increase in bone mineral density [154]. Pre-menopausal women suffering from polycystic ovarian symptom (PCOS) also saw improvements in their symptoms, including better insulin resistance [155]. In pregnant women, there was a reported increase in depression when isoflavones were consumed in high quantities [156]. Also, women who had no previous history of hormonal imbalances, reported increased discomfort and prolonged menstruation [157].

Infants were also shown to be sensitive to isoflavones. As soy has become more commonly used as a high-protein formula replacement for infants, concerns have been raised on the increased isoflavone consumption at young ages. It is reported that high intakes of phytoestrogen during the mini puberty phase of infant development can lead to disfigurements of the reproductive organs [158]. In males, this was observed as malformation of previously normal testes, and in female infants, the maturation of vaginal cells at as young as 6 months was observed. Both male and female infants and toddlers were reported to have an increase in breast tissue when compared their peers [157].

There is also evidence to support that the consumption of the isoflavone molecule GEN is beneficial as a treatment and management of cardiovascular diseases, such as cholesterol levels and high blood pressure [159,160]. One study concluded that increasing daily soy consumption by 50 g decreased cholesterol by an average of 3% [161]. There has even been a study that found isoflavones from multiple pea sources had some anti-proliferative effects on tumor cells [162].

The reports of effects on hormonal cancers are mixed with some studies reporting a positive reduction in risk factors for breast and prostate cancer, while others report an increase in tumor growth [163,164]. This pattern of both positive and negative health effects appears to be common. Because phytoestrogen mimics the hormone estrogen, the effect of increased consumption seems to rely on an individual's sex, age, and reproductive stage of life.

There are pros and cons of isoflavones for animal/livestock health as well. The increased consumption of soy by livestock is beneficial as it leads to increased weight gain, a desirable trait among producers [165]. A study conducted on increased isoflavones in swine diet concluded that isoflavones were positively correlated with growth performance, improved antioxidative properties, and the protection of intestinal morphology [166]. A similar study conducted with poultry concluded the same benefits witnessed in swine [167]. In dairy cows, increased consumption of red clover, which is rich in isoflavones, was shown to increase the overall milk production as well as anti-inflammatory and immune factors [168].

Similar to humans, the effects of increased isoflavone content in animals vary depending on sex and reproductive stage. Consuming toxic levels of isoflavones during early growth stages can lead to increased fertility issues in some animals [169]. In dairy calves, increased isoflavones negatively impacted the reproductive health at maturity [170]. Overconsumption in males demonstrated a negative effect on testes development, leading to a decrease in fertility rates [171]. These results seem to be consistent across cattle, swine, sheep, and poultry [169,171]. All these effects can be seen in Figure 1.

## EFFECT OF INCREASED ISOFLAVONE CONSUMPTION ON HEALTH BY SPECIES

### POSITIVE

- improve cardiovascular health
- reduce menopausal symptoms
- reduce risk of hormonal cancers
- reduce PCOS symptoms
- increase bone density

- increase growth performance in cattle
- increase milk production
- increase immune factors and anti-inflammatory capacity

- increase growth performance
- improve antioxidant properties
- increase protection of intestinal morphology

- increase growth performance
- improve antioxidant properties
- increase protection of intestinal morphology
- increase egg production

### NEGATIVE

- increase depression among pregnant women
- prolong menstrual cycles
- malformation of reproductive organs in infants

- malformation of reproductive organs in calves
- decrease fertility rates

- negatively impact teste formation
- decrease fertility rates

- negatively impact teste formation
- decrease fertility rates

**Figure 1.** Comparison of the positive and negative impacts of increased isoflavone consumption in humans, cattle, swine, and poultry species [157–161,163–165,167–171].

Given the many studied impacts of isoflavones on human and animal health, there is significant promise for future use. For humans, it could be isolated and utilized in prescription medication to reduce the symptoms associated with menopause and heart diseases. Regarding animals, its use as a dietary supplement could be used to increase weight gain for both the benefit of producers and consumers. However, it should be approached cautiously with continued research efforts to fully understand the impact of isoflavones in different species and subgroups of the population.

## 5. Antinutritional Factors

### 5.1. Antinutritional Factors Importance and Overview

Legumes, specifically soybeans, serve as significant sources of plant protein for animals and humans. However, legumes' high protein content also comes with several antinutritional factors. Examples include trypsin inhibitor (TI) and lectin, which both render the digestion of nutritional components challenging and limit the nutritional impact of legume seeds.

Recently, there has been a resurgence of interest in the antinutritional factors in legumes. TIs are found in all plant tissues and play an important role in a plant's defense against feeding from pests as proteins that bind strongly to trypsin, a digestive enzyme of the pancreas, block its active site [172,173]. TIs are comprised of two main polypeptides–KTI and BBI, constituting approximately 6% of the protein present in soybean seeds. A comparison of the two TIs can be seen in Figure 2. The aspartic, serine, and cysteine proteases are targeted by KTI, while BBI targets trypsin, chymotrypsin, and elastase [174]. Soybean, pea, mung bean, and other legumes, despite being an important source of protein for humans and animals, cannot be consumed raw because of their high TI concentrations, with soybean having the highest [175].

| | KTI | BBI |
|---|---|---|
| Protein structure | N-ter | N-ter |
| Protein molecular weight | 18-22 kDa | 8-16 kDa |
| Target proteases | Trypsin | Trypsin and chymotrypsin |
| Heat resistance | Remain 30% activity with 100°C for 5 minutes | Remain 89% activity with 100°C for 5 minutes |
| Number of amino acids | 150-240 amino acids | 54-113 amino acids |
| Isoelectric point | 4.2 – 9.6 | 4.0 – 9.8 |

**Figure 2.** Comparison of KTI and BBI [176]. The protein 3D structures of Glyma.08g341500 and Glyma.16208900 (wm82.a4.v1) were generated by the AlphaFold protein structure database (https://alphafold.ebi.ac.uk/) (accessed on 8 March 2024).

The literature shows that lectin is a natural bioactive protein and glycoprotein compound that possesses the unique capability to specifically bind sugars [177–181]. Originating from non-immune sources, these sugar-binding proteins can agglutinate cells or precipitate glycoconjugates [182,183]. A recent study concluded that lectins serve as a valuable model for studying protein–carbohydrate interactions and as a precise tool for analyzing carbohydrate-bound proteins (free form, lipid, or protein) [184]. Numerous studies have argued about the distinctive characteristics of both plant and animal lectins as critical identification molecules in cell–cell and cell–molecule interactions across biological systems [185]. Additionally, lectins contribute substantially to clarifying the structure of carbohydrates, biological processes, and clinical diagnostic systems [185,186]. Additionally, lectins can perturb overall nutrient metabolism, leading to enlargement or organ degeneration, and altering the hormonal and immune response [187]. While food allergens typically resist gut digestion, lectins, although stable, generally exhibit lower allergenicity compared to other allergens [188]. Given this, lectins are recognized as minor allergens in soybeans [189]. For example, purified lectins extracted from soybeans prevent rat growth, induce intestinal and pancreatic hypertrophy, and hyperplasia of the pancreas. These lectins prevent nutrient assimilation by attaching to intestinal epithelial cells, concurrently leading to intestinal tract damage, and facilitating bacterial access to the bloodstream [188].

The importance of lectins is well documented, with plant lectins being predominantly abundant in seeds. Despite this, their presence extends across various vegetative tissues, including flowers, leaves, roots, barks, rhizomes, and bulbs [190,191]. Lectins have been shown to directly engage with the intestinal epithelium, which can result in the interruption of nutrient absorption and transportation [192,193]. Additionally, previous studies have emphasized that lectin-containing diets are associated with changes in gut immune responses, decreased gut hormones, and mucosal cell damage [194]. The harmful effects of lectins are marked by growth inhibition in experimental animals and the onset of symptoms such as diarrhea, nausea, bloating, and vomiting in humans [195].

*5.2. Comparison of Antinutritional Factors across Species*

BBIs are extensively studied serine protease inhibitors found abundantly in both dicotyledonous and monocotyledonous plants, though recent work has revealed distinct groupings for BBIs between monocots and dicots [196]. The overall topology of BBIs indicates a divergent evolutionary pattern for each group. BBIs from dicots exhibit significant conservation, with minor evolutionary variations observed, while those from monocots display considerable variability, suggesting an intriguing evolutionary process driven by internal gene duplications and mutation events. Analysis of dicot structure features showed that BBIs typically possess a molecular weight of 8 kDa and feature a double-headed structure with two reactive sites. Conversely, in monocots, BBIs can be categorized into the following two classes: one approximately 8 kDa with a single reactive site (having lost one reactive site), and the other approximately 16 kDa with two reactive sites [197–199]. These reactive sites are situated on unique exposed surfaces formed by a disulfide-linked β-sheet loop, which is highly conserved, rigid, and primarily composed of nine residues. This finding implies that gene duplication events play a pivotal role in molecular evolution [200]. Despite alterations in the amino acid composition of BBIs during evolution, their cysteine residues remain highly conserved.

Unlike BBIs, KTIs appear to have diverse functions. Its expression pattern varies among different tissue types. Soybean KTI1, KTI2, and KTI3 genes were reported to be exclusively expressed in seed tissue [201]. This finding has been confirmed by an additional study, and it was further demonstrated that some soybean KTI genes are also expressed in vegetable tissues [202]. Similar findings were observed in other plant species, suggesting potential roles of KTIs in plant development and defense responses in vegetable tissues. For example, at least two of the KTI gene products strongly inhibited proteases in the midgut extracts of *Malacosoma disstria*, a lepidopteran pest of Populus [203]. Arabidopsis lines containing silenced KTI genes, atkti4 and atkti5, were found to be more susceptible to *T. urticae* (Spider mite) than wild-type plants [204]. In alfalfa, the TI proteins Msti-94 and Msti-16 were shown to act as stomach poisons, significantly reducing the survival and reproduction rates of aphids [205]. In wheat, α-amylase/trypsin inhibitor (ATI) CM3 was identified as pest-resistance molecules, activating innate immune responses in monocytes, macrophages, and dendritic cells [206].

Certain lectins exhibit detrimental effects on the gastrointestinal system by damaging intestinal epithelia, impeding nutrient assimilation, and causing alterations in the microbiota [188]. While some of the effects of lectin are known, there remains a limited amount of research regarding lectin effects, with most of the research only being recently conducted. In terms of activity between pea, chickpea, and soybean, soybean has a much higher lectin activity. Specifically, the lectin activity in pea is around 5.64 hemaglutinin activity (HU)/mg, chickpea is around 2.74 HU/mg dry flour, and soybean has lectin activity of around 692.8 HU/mg dry flour [207]. Prior research suggests that soybean, chickpea, faba bean (*Vicia faba*), pea, and others are notably recognized as sources of lectins [195]. In terms of content, usually faba beans contains a lower content when compared to peas, although these two crops show significantly low content compared to soybeans meal [195]. Data from several studies have identified that a central characteristic of lectins comes from their ability to impede the uptake of nutrients in the small intestine area [195]. There exists a considerable body of literature on lectin types, and the classification scheme (Figure 3) delineates various categories based on distinct attributes, including the following: overall structure, structurally and evolutionary related proteins, and affinity towards specific carbohydrate moieties [184]. Under the taxonomy of carbohydrate structure, lectins are classified as merolectins (possessing singular carbohydrate binding sites, which herein is the only carbohydrate-binding domain in this protein [208]), hololectins (exhibit a minimum of two identical and similar carbohydrate-binding domains), superlectins (featuring a combination of two non-identical carbohydrate-binding domains that will attach to different sugars), and chimerolectins (due to the absence of the carbohydrate domain itself, the chimeric protein holds off on the carbohydrate binding and tags to another domain) [8]. Additionally, lectins are grouped according to their structurally and evolutionary related proteins, encompassing families such as Amaranthin (seeds from *Amaranthus*), chitin binding, Cucurbitaceae phloem, jacalin related (seeds of the jack fruit (*Artocarpus integrifolia*)), legumes, monocot mannose binding (found in monocotyledonous plants), and type 2 ribose inactivating lectins (catalytically inhibitor of eukaryotic ribosome [209–213]. Moreover, lectins are further categorized based on their binding affinity towards specific carbohydrate moieties, including glucose, galactose, and N-acetyl β-galactosamine, L-fucose, and sialic acid [184]. This systematic approach to lectin classification provides a rigorous framework for understanding the structural diversity, evolutionary relationships, and carbohydrate-binding specificity within the lectin family.

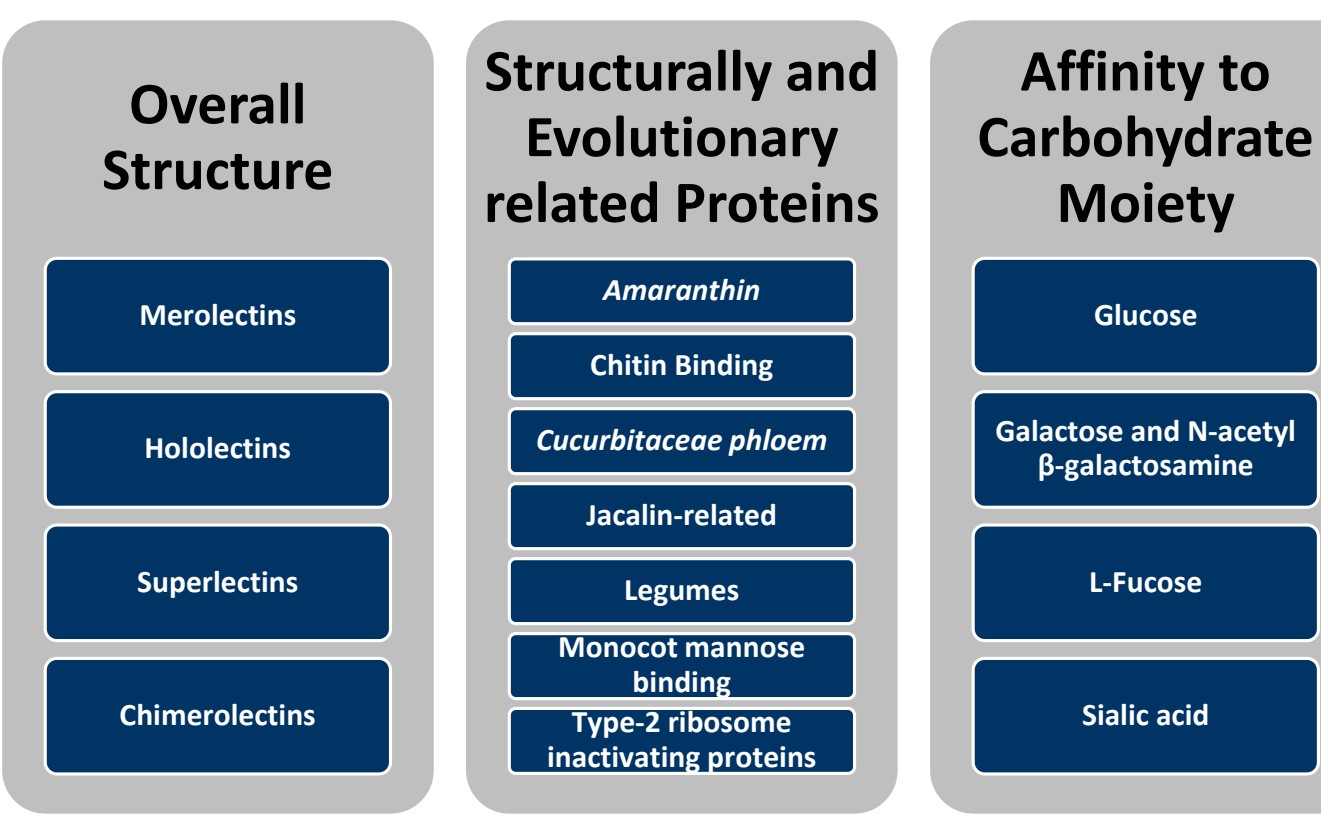

**Figure 3.** Classification of plant lectins [184].

*5.3. Uses of Antinutritional Factors in Research and Future Directions*

TIs can cause significant health issues for both animals and humans. They hinder the activity of pancreatic serine proteases, thereby disrupting plant protein digestion [213]. This restriction results in the reduced absorption of protein and essential nutrient absorption from legumes [213]. This can limit livestock growth and, in severe cases, contribute to nutrient deficiencies as well as the enlargement of the pancreas, liver, intestines, and pancreatitis [213].

In order to make plants containing high levels of TI consumable, it must be treated with heat. A heat treatment denatures the TI, making it more digestible for both humans and animals. Studies report that BBI exhibits a higher tolerance to heat treatment than KTI [214]. In the case of soymilk, heating at 100 °C for 15 min causes KTI to readily form protein aggregates through noncovalent or disulfide bonds, resulting in a 70% reduction in its trypsin inhibitory activity. Conversely, approximately 89% of chymotrypsin inhibitor activity (CIA), primarily attributed to BBI, can remain after heating at 100 °C for 15 min [215]. Therefore, food processing facilities heat seeds to temperatures exceeding 108 °C for a duration of 15 min to deactivate both major TI proteins [216]. This, however, leads to a loss of nutritional factors within the seed as well [217].

However, it has been reported there are some benefits of TI regarding human health. TI has been suggested as a treatment for obesity and metabolic disorders due to their modulation of satiety hormones [218]. Specifically, in human and mice trials with TI treatments isolated from peanut paçoca showed reduced food consumption and even caused weight loss, with no signs of pancreatic toxicity. Furthermore, a trypsin inhibitor isolated from *Tamarindus indica* L. (TTI) was able to reduce the food intake of eutrophic Wistar rats by about 47% [219]. In addition to controlling weight, soybean KTI has been studied as a potential treatment for inflammatory lung diseases [220]. In an in vivo mouse model in which lipopolysaccharides (LPS) from bacteria induced acute lung injury in the mice, purified soybean KTI was able to significantly suppressed the inflammatory effects caused by elastase in a dose-dependent manner [220]. KTI has also been found to have some

anticancer properties. Soybean KTI has been found to inhibit ovarian cancer cell growth, while chickpea KTI has been found to inhibit both breast and prostate cancer cell proliferation [17,221]. BBI has been extensively reported to possess functional anticancer properties. It is considered as a drug by the FDA and has been shown to reduce the risk of heart disease and breast cancer [222,223]. In vitro studies utilizing BBI have demonstrated its effectiveness as an anticarcinogen even at nanomolar concentrations, with irreversible effects on cancer cells [222,224]. Moreover, BBI has been observed to reduce the size of precancerous lesions in the mouth, known as leukoplakia, in about one-third of participants [225]. Due to TIs' ability to combat some cancers and decrease heart disease, more research is expected to elucidate the mechanism. Research will also be necessary to understand why BBIs have more significant anticancer effects than KTIs. As TIs have been able to combat cancers in human cells, it is possible for studies to determine if they can also combat cancers in other animal species.

TIs have also been studied for its potential as a biological pesticide, due to its plant tissue function in the defense against insect pests. Purified and concentrated TIs from seeds have been shown to create an effective bio control for melon fruit flies as well as increase the resistance and performance of *Bt* corn [226,227]. Additionally, TIs from soybean, chickpea, and pea have all been found to control a variety of insects including the larvae of *Spodoptera litura*, the cotton boll weevil (*Anthonomus grandis*), and pea aphids (*Acyrthosiphon pisum*) [228–231]. These findings hold promise for the development of more sustainable biocontrol methods. Research exploring TI as a biological pesticide should continue as the need for more environmentally sustainable pesticides increases. This type of research would gather more attention due to the importance of finding pesticides that can successfully deter crop pests without causing significant damage to the environment.

Considering the reported plant lectin activities for human health, it is conceivable that clinical interest in certain lectins will increase due to its many benefits, including anti-inflammatory and antihemolytic effects, as well as potential for healing cutaneous wounds [232,233]. Recent research shows that lectin exhibits promising control over tumor cell metastasis by inducing programmed cell death. This potentially offers anti-inflammatory properties via the lectin domain in legumes, and demonstrates immunomodulatory effects [234–238]. Additionally, these proteins demonstrate antifungal action and generation of cytokines in both in vitro and in vivo [239–241]. Pea lectin was specifically found to trigger apoptosis in some human colon cancer cells, while chickpea lectin was able to cause apoptosis of human breast cancer [242,243]. Interestingly, chickpea lectin has not only been found to have anticancer effects, but also antiviral properties by inhibiting HIV-1 reverse transcriptase [244]. Lectin has also been found to be an optimal candidate for treating infections caused by pathogens, as they can specifically bind to and block chitin-containing pathogenic bacteria, inhibiting microbial adhesion [245,246]. Given these health benefits, lectin research focusing on anticancer and antiviral properties will continue. A study performing a direct comparison between the anticancer properties of lectins between different legumes to determine if the lectin from one legume is more effective at inhibiting cancer than others would be of particular interest. Based on current research, chickpea lectin seems to be the most effective, but more data are necessary to determine this.

While lectins seem to have positive effects in humans, their impact within other species is not well understood. In fish, such as Rainbow Trout (*Oncorhynchus mykiss*), some cases found that lectin caused negative reactions in the gut health while others found no negative or positive impact [247,248]. In juvenile rats fed diets with increased lectin content, they experienced a decrease in growth by 20% compared to the control [249]. Additionally, both rats and pigs were found to have some adverse effects from lectin, such as inflamed stomach and pancreas [187,250]. Given these negative effects from lectin, undoubtedly more research will have to be conducted to fully determine when lectin can have beneficial or negative effects on animals.

Finally, another promising research line recognizes plant lectins as natural endogenous protective substances against herbivores, imposing harmful impacts on the gut system [251].

The ingestion of lectins by insect larvae leads to various detrimental outcomes, including restriction of growth, decrease in grain size and weight gain, disruption of female fecundity, decreased proportion of adult emergence and pupal development, and prolonged developmental duration, ultimately resulting in larval mortality. These results are significant because lectins are proposed as promising agents for insect pest management and have been effectively engineered into crops [252].

## 6. Saponins

### 6.1. Saponins Importance and Overview

Most early studies, as well as current research, focus on the importance of soyasaponins, the most common saponins encountered in legume plants. The structure of saponins are made up of an aglycone (a chemical compound that is a non-saccharide) and oligosaccharide components [253]. For saponins, most aglycones are sapogenins and soyasapogenols, which are both steroids. These compounds are typically categorized into subgroups, and they can be distinguished by their aglycone structures. The subgroups are divided into soyasaponin groups A–E, representing the glycosides of soyasapogenols A–E, respectively, and group DDMP (2,3-dihydro-2,5-dihydroxy-6-methyl-4H-pyran-4-one) soyasaponins are characterized as glycosides of soyasapogenol B, incorporating C-22 chains bound to DDMP residues [253]. Aside from legume seeds, saponins are also found to accumulate in various plant organs, including leaves, tubers, nodules, flowers, and fruits [254,255]. Saponins present in soybeans, constituting approximately 0.5% of dry matter, primarily reside in seed hypocotyls rather than cotyledons. However, the saponin content is subject to significant variation depending on cultivars, maturity levels, and growth locations [256,257].

Previous reviews have elucidated that saponins offer numerous health benefits, including the reduction of blood lipids, mitigation of cancer risks, modulation of blood glucose response, and preventing of platelet aggregation [258]. Some authors have also suggested that the therapeutic potential extends to alleviating hypercalciuria and serving as an antidote against acute lead poisoning [6]. Additionally, saponins exhibit expectorant and antitussive properties [259]. It is important to note that despite the numerous beneficial effects from saponins, there are studies that highlight the limitations of these compounds, such as their propensity to form insoluble complexes with proteins, lipids, and essential minerals such as iron, zinc, and calcium, ultimately impeding nutrient absorption in the body [260].

### 6.2. Comparison of Saponins across Species

Recent investigations continue to elucidate the content and bioactivity of saponins among legumes. Table 2 presents the content of saponins in various legume species as a percentage of dry weight and gram per kg of dry matter. Chickpeas exhibit a relatively wide range of saponin content, spanning from 0.26% to 6% of dry weight and 2.3 g/kg of dry matter. Soybeans display a comparatively narrower range, with saponin content ranging from 0.5% to 2.5% of dry weight and about 20 g/kg of dry matter. Peas demonstrate lower saponin content, ranging from 0.01% to 0.18% of dry weight and 1.8 g/kg of dry matter, while mung beans exhibit the lowest saponin content among the plant-based proteins studied, with values ranging from 0.05% to 0.057% of dry weight and 0.5 g/kg of dry matter. These findings clarify the diversity in saponin content among diverse legumes species, offering valuable insights about their nutritional profile and potential health effects.

There is also documentation of bioactive properties associated with various plant-based protein species (Table 2) [261]. Chickpeas exhibit antimicrobial activity, indicating their potential role in inhibiting the growth of microorganisms as well as having some preventative effects in a wide range of human health issues, including diabetes and heart disease [262,263]. Soybeans have been shown to reduce growth performance and feed efficiency in fish, while also demonstrating antioxidant activities and decreased blood pressure [264,265]. However, soybeans may also induce intestinal inflammation, highlight-

ing potential adverse health effects for animals and humans [266]. Contrarily, saponins from soybeans have been found to prevent the proliferation of some human cancers and inhibit HIV infections [267,268]. Peas, meanwhile, demonstrate the inhibition of digestive enzymes, suggesting a potential role in nutrient absorption [269]. An inhibitory effect against digestive enzymes such as pancreatic lipase and α-glycosidase was also reported as a potential health benefit [269]. The least is known about saponins in mung beans, due to their low content. Albeit, some saponins in mung beans have been noted for their antioxidant abilities as well as anti-proliferative effects on certain human cancer cells by blocking cell cycle progression [270,271]. Overall, these findings highlight the diverse bioactive properties of plant-based protein, which may have implications for human health and disease prevention.

**Table 2.** Saponin content and bioactive properties in plant-based protein species [261,266,268,270–272].

| Species | Common Name | Saponin Content (g/kg of Dry Matter) | Concentration (% Dry Weight) | Bioactive Properties |
|---------|-------------|--------------------------------------|------------------------------|----------------------|
| *Vigna mungo* | Mung bean | 0.5 | 0.05 to 0.57% | Anticancer, antioxidant |
| *Pisum sativum* | Pea | 1.8 | 0.01 to 0.18% | Suppression of digestive enzymes (pancreatic lipase and α-glycosidase) |
| *Cicer arietinum* | Chickpea | ~20 | 0.26 to 6.0% | Antimicrobial, antidiabetic |
| *Glycine max* | Soybean | 6.5 | 0.5 to 2.5% | Lower growth performance and digestion, antioxidant, reduces blood pressure, cause intestinal inflammation, anticancer, antiviral |

### 6.3. Uses of Saponins in Research and Future Directions

Due to the reported human health benefits, saponins find extensive application in the cosmetic industry as natural emulsifiers, foaming agents, and cleansing agents. Saponins enhance the formulation of these various personal care products by improving their lathering, cleansing, and moisturizing capabilities [273]. Additionally, high concentrations of saponins have been identified as potential natural rumen manipulators, capable of influencing the composition and fermentation patterns of ruminal microbial populations. They influence microbial composition through ruminal defaunation, where they suppress ciliate protozoa and consequently enhance the efficiency of microbial protein synthesis by reducing microbial protein turnover and duodenal protein flow. Furthermore, saponins have been observed to impact ammonia adsorption and modulate the passage of digesta in the rumen, leading to alterations in ruminal metabolism with minimal physiological responses compared to microbiological effects [274].

The variety of health benefits for saponins position researchers well for increased medicinal research studies. More research is needed to determine saponin mechanisms for beneficial health effects. Even with the medicinal properties of saponins, it would be surprising for legumes to be specifically grown for medicinal saponin production due to just how little saponin content is naturally in legumes. In order to obtain sufficient saponin to produce enough for medical products, legumes would need to have drastically increased saponin levels, which is not possible through traditional breeding. As such, if more research is performed for saponin medicinal usage, it will likely involve synthetically made saponins [275]. Synthetically made saponins are easier to produce due to low biological saponin content as well as microheterogeneity and laborious extraction methods [275].

Given the widespread usage of legumes in animal feed, it is unsurprising that there has been sufficient research focusing on the impact of saponins on animal growth. The exact growth effects of saponins seem to be highly dependent on the species. In fish, it has mostly a negative effect, as reported in Rainbow trout, Chinook salmon (*Oncorhynchus tshawytscha*), and Atlantic salmon (*Salmo salor*, L.), with decreased growth, feed intake, and moderate

intestinal damage in Rainbow trout [276,277]. In European Sea bass (*Dicentrarchus labrax)*, saponins did not have a significant effect on growth but still caused some minor digestive issues [278]. While saponins tend to negatively affect fish, some researchers have found positive effects in mammals fed with saponins. Some cases found that diets enriched with saponins increased milk production in cows, increased wool production in sheep, and decreased blood cholesterol levels in sheep [274,279–281].

Due to saponins having varying effects on animal growth and feed, future research goals involving saponins should seek to modify the levels in crops through breeding. Thus, breeders could develop legume varieties with different saponin levels depending on the animal species which will be consuming them. If the particular crop is being grown for fish feed, breeders could work to develop low saponin lines, while if it is being grown for cows or sheep, breeders could develop high saponin lines. While this type of research has rarely been performed in legumes for saponins, it has been successful in alfalfa for saponin levels, justifying the case for more traditional breeding [282]. Of legumes, soybean and chickpea would be the most likely targets for modified saponin content due to limited contents in others. Besides breeding, saponin levels could be changed through post-harvest methods. Current methods, such as soaking and blanching, are known to lower saponin levels by facilitating dissolving in water and removal [258]. This could facilitate the removal of saponins in legumes prior to feeding animals that have negative reactions to saponins, such as fish. While soaking and blanching can lower saponin levels, more research can be carried out to improve and develop more efficient methods at lowering saponin levels.

## 7. Conclusions

In summarizing the vast expanse of research covered in this review, it is evident that legumes serve as a reservoir of bioactive compounds with profound implications for both human and animal health. Soybeans, peas, chickpeas, and mung beans as primary legume protein sources harbor a diversity of peptides, protein subunits, isoflavones, antinutritional factors, and saponins. Each of these bioactive compounds contributes uniquely to the nutraceutical potential of legumes, offering benefits ranging from antioxidant and anticancer activities to the management of chronic diseases such as diabetes and hypertension.

Notably, this review underscores soybeans' unparalleled bioactive compound profile, which sets a benchmark for nutritional and functional research. However, the potential of other legumes like chickpeas and mung beans, despite being less explored, suggests a vast, untapped resource for health-promoting bioactive compounds. The comparative analysis of these compounds across different legumes provides a foundation for future studies aimed at enhancing our understanding of their health benefits and mechanisms of action.

Future research should pivot towards not only elucidating the detailed mechanisms through which these compounds exert their effects, but also exploring the synergistic relationships between them. Additionally, there is a pressing need for clinical trials to validate the health claims associated with legume-derived bioactive compounds and to determine their efficacy and safety in human populations. The exploration of genetic and agronomic strategies to enhance the bioactive compound content in legumes could also pave the way for the development of functional foods tailored for specific health outcomes.

Moreover, as this review has highlighted, the application of legume bioactive compounds extends beyond human health, impacting animal nutrition and environmental sustainability. The potential of legumes to serve as a sustainable protein source, coupled with their bioactive compounds, positions them as a key player in addressing the global challenges of food security, nutrition, and climate change.

In conclusion, this review comprehensively underscores the crucial role of legumes as a source of health-promoting compounds and paves the way for future research to fully exploit their potential. Moving forward, it is crucial that interdisciplinary research spanning food science, nutrition, agronomy, and pharmacology unite to harness the nutraceutical benefits of legumes, aiming to improve human health and environmental sustainability.

**Author Contributions:** Z.S. was involved with the conceptualization, writing—original draft preparation, and editing. M.O.d.G. was involved with the writing—original draft preparation. E.B.F. was involved with the writing—original draft preparation. Y.Z. was involved with the writing—original draft preparation. P.B. was involved with the writing—original draft preparation. Z.W. was involved with the writing—revision and editing. W.M.S. was involved with the writing—revision and editing. B.Z. was involved with the conceptualization and writing—revision and editing. All authors have read and agreed to the published version of the manuscript.

**Funding:** This research received no external funding.

**Acknowledgments:** We would like to acknowledge the USDA-ARS, Virginia Tech soybean breeding lab, and the Donal Danforth Center for their scientists' help in writing this review.

**Conflicts of Interest:** The authors declare no conflicts of interest.

**EEO/Non-Discrimination Statement:** The U.S. Department of Agriculture (USDA) prohibits discrimination in all its programs and activities on the basis of race, color, national origin, age, disability, and where applicable, sex, marital status, familial status, parental status, religion, sexual orientation, genetic information, political beliefs, reprisal, or because all or part of an individual's income is derived from any public assistance program. (Not all prohibited bases apply to all programs.) Persons with disabilities who require alternative means for communication of program information (Braille, large print, audiotape, etc.) should contact USDA's TARGET Center at (202) 720-2600 (voice and TDD). To file a complaint of discrimination, write to USDA, Director, Office of Civil Rights, 1400 Independence Avenue, S.W., Washington, D.C. 20250-9410, or call (800) 795-3272 (voice) or (202) 720-6382 (TDD). USDA is an equal opportunity provider and employer.

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
