# Peer review of "A Review of Bioactive Compound Effects from Primary Legume Protein Sources in Human and Animal Health"

_cimb, doi:10.3390/cimb46050257_

Round 1

Reviewer 1 Report

Comments and Suggestions for Authors

1. Abstract can be more informative and constructive.

2. In Introduction, about bioactive compounds author can write in single paragraph.

3. Author should elaborate research gap and novelty of this study in Introduction section.

4. Few relevant recent references should be added to strengthen this Ms. (https://doi.org/10.3390/foods12213898, https://doi.org/10.1007/s12033-023-00937-2)

5. Author can improve the quality of Figures.

6. Heading 5 and 6 both are same "Conclusions" author should check and correct accordingly.

7. References should be as per journal guidelines and cross checked carefully.

Comments on the Quality of English Language

Typological mistakes should be thoroughly checked in Ms

Reviewer 2 Report

Comments and Suggestions for Authors

The topic of the review is interesting. However, we need to admit that the review topic is not novel and there are several recent review articles that have comprehensively discussed this topic. Some examples:

Full article: Plant-derived proteins as a sustainable source of bioactive peptides: recent research updates on emerging production methods, bioactivities, and potential application (tandfonline.com)

Full article: Recent Development in Bioactive Peptides from Plant and Animal Products and Their Impact on the Human Health (tandfonline.com)

Antioxidants | Free Full-Text | Health Benefits of Antioxidative Peptides Derived from Legume Proteins with a High Amino Acid Score (mdpi.com)

Bioactive peptides derived from plant origin by-products: Biological activities and techno-functional utilizations in food developments – A review - ScienceDirect

Review on plant-derived bioactive peptides: biological activities, mechanism of action and utilizations in food development - ScienceDirect

I don't think that this work will contribute to the existing literature. In fact, the content is not well-organised and I don't see a good flow of ideas/structure. For example, section 5, which is titled conclusion, comes between two sections related to isoflavones and saponins. Nor does the content of section 5 represent the conclusion and nor does the content fit between those two sections. Furthermore, when the authors speak about the uses/benefits in different sections, they are briefly summarising the basic literature based on selected studies which does not match the aim of their review ("comprehensive review"). There are several other weaknesses too.

Overall, I would recommend rejecting this manuscript.       

Reviewer 3 Report

Comments and Suggestions for Authors

Dear authors,

Please, I attach the manuscript with the suggested changes. Please note the corrections made in the document, everything is underlined and commented by the reviewer.

Aditionally, I emphasize the following suggestions:

-Please reformulate the conclusion, it is a little bit poor.

-I urge authors to use more images as it is a long revision and makes it more enjoyable to read.

Comments on the Quality of English Language

The use of English needs to improve. There are sentences that are not well expressed and are difficult to understand for the reader.  I urge the authors to check the English throughout the text. 

Reviewer 4 Report

Comments and Suggestions for Authors

This article provides a comprehensive overview of bioactive compounds in two primary legumes, soy and peas, and their potential health benefits for humans and animals. Although the information is valuable, the article lacks an essential element of novelty. It could be strengthened by incorporating new research or innovative approaches to the topic. While two primary legumes are mentioned in the title, the article focuses primarily on soybeans. More consideration should be given to peas and other legumes to maximize their potential. The article could benefit from more specific data and scientific evidence to support its claims regarding the health benefits of the compounds discussed. Based on this assessment, the article has the potential to be a valuable contribution to the journal “Curr. Issues Mol. Biol.” with some revisions. By addressing the areas for improvement and further strengthening the article's focus, innovation, and evidence base, authors can increase its chances of publication.

·         Introduction – The title mentions legumes, but the introduction focuses on the general plant protein market. Refine the introduction to highlight legumes (soy and peas) as the main focus. Various benefits and bioactive compounds are mentioned in the introduction. Since the title focuses on two legume sources, consider narrowing down the health benefits and bioactive compounds discussed to those most relevant to soy and peas. The connection between bioactive compounds and health benefits needs to be stronger. Briefly mention how these compounds contribute to the overall health benefits of soy and pea protein. Although the market size is interesting, it can be condensed or even removed because it has no direct connection to the core topic.

·         2. Bioactive Peptides – It might be helpful to provide a brief introduction or overview at the beginning of the section to provide context for the discussion of bioactive peptides from legumes. Provide additional explanations or examples to support the statements. For example, when discussing the functions of bioactive peptides, elaborate on how they exert antihypertensive, antioxidant, antimicrobial, antidiabetic, and hypocholesterolemic effects. Provide more clarity on the process of mass production of bioactive peptides and the various methods involved such as enzymatic hydrolysis and microbial fermentation. Expand the discussion on bioactive peptides from legume sources other than soybeans. While soybeans are covered in detail, providing further insight into peptides from other legumes such as chickpeas, peas, mung beans and cowpeas would enrich the review. Incorporate additional research or case studies to illustrate the effectiveness of bioactive peptides from various legume sources on human and animal health. Provide more specific references to studies that support the discussed functions and benefits of bioactive peptides, particularly those from non-soy legume sources.

·         3. Protein Subunits – The introduction (3.1. Protein Subunits Importance and Overview) overlaps with information from the abstract and introduction of the article. This paragraph focuses on the benefits of soy protein, but does not directly refer to protein subunits. You might consider mentioning how the subunits contribute to these health benefits. You can summarize this section by focusing solely on the importance of protein subunits in legumes. Some sentences are long and complex. Consider breaking these sentences down to improve readability. While subunits from other legumes (lentils, peas, mung beans) are mentioned in the section, the focus is primarily on soy protein. You can title “3. Protein subunits” to “3. Soy Protein Subunits and Comparisons Between Legumes”. Discuss possible applications of subunit ratio adjustments.

·         4. Isoflavones – Briefly mention that isoflavones may be present in different concentrations depending on the processing method used for legume protein sources. Briefly mention how isoflavones might interact with or be affected by the protein content or composition of legumes. Consider adding a table summarizing the main human and animal health effects of isoflavones (positive and negative). You may wish to mention any limitations of current research on isoflavones.

·         5. Conclusions – The section should be revised to ensure that it directly addresses the effects of antinutritive factors from legume protein sources on human and animal health, consistent with the title of the article. Ensure that the conclusions drawn are clearly stated and logically organized to provide a coherent summary of the nutritional inhibitory factors discussed. Provide a more balanced discussion by including any potential benefits or positive aspects of nutritional factors affecting human and animal health. Expand the discussion of lectins to cover their diverse health effects, including potential therapeutic uses and future research directions. Incorporate additional research or case studies to support conclusions regarding the health effects of anti-nutritional factors and their potential applications.

·         6. Saponins – Consider a title that reflects the focus on human and animal health, e.g., "6.1 Saponins: Bioactive compounds with potential health benefits in humans and animals”. While the section mentions animal health, the focus is on human benefits. Balance the discussion by adding more details about animal-specific health effects of saponins (positive or negative). Explain the “future directions” of saponin research, particularly as they relate to human and animal health. The section focuses heavily on soybeans. If the article is about two primary sources of legumes, make sure the other source receives equal attention. You might consider mentioning possible methods for reducing saponin levels in legumes, especially if these are relevant to the protein sources chosen. Briefly discuss possible future research directions related to saponins in legumes and their effects on human and animal health. Minor grammatical errors throughout the section.

·         7. Conclusion – Connect this section with Section 5. The conclusion should specifically address the two primary legume protein sources (mentioned in the title) and their bioactive compounds. While comparisons to other legumes are fine, prioritize those that are relevant to the core topic of the article. Instead of listing all the bioactive compounds, consider highlighting the most important ones that have been discussed in the context of the two primary legumes. Explain the effects of these bioactive compounds on human and animal health. Review the main points of the article. Briefly mention the mechanisms by which they might influence certain health conditions. Briefly emphasize the importance of this research in promoting the use of these legumes as protein sources and their contribution to human and animal health. Briefly discuss potential future research areas related to these bioactive compounds and their role in promoting human and animal health.

·         Abstract – The summary should clearly identify the two primary legume protein sources that the article focuses on. Instead of listing all the compounds discussed, highlight the most relevant to the legumes you selected. Mention specific examples relevant to the compounds studied. Briefly mention the importance of comparing these compounds between legumes. Briefly mention the potential applications of this research, e.g. promoting the use of legumes due to their bioactive compounds and health benefits. Avoid vague "many health benefits".

Comments on the Quality of English Language

Minor editing of English language required. 

Reviewer 5 Report

Comments and Suggestions for Authors

Comments on the Quality of English Language

Moderate editing of English language required.

Round 2

Reviewer 2 Report

Comments and Suggestions for Authors

The authors have improved the manuscript, however, I am not convinced about the novelty of the manuscript considering the number of current recent literature reviews that have "comprehensively" considered this topic. 

It is up to the editor to decide on this, however, if the editor wishes to accept it, then the term "Comprehensive" needs to be removed from the title because many of the aspects are discussed "generally" with "selected" studies, rather than having a comprehensive approach (example: when discussing the uses/benefits of the different bioactives). 

Reviewer 4 Report

Comments and Suggestions for Authors

The manuscript has been effectively revised and enhanced by the authors. All my suggestions were well taken into account by the authors. Therefore, this manuscript may be considered for publication in this journal.

Author Response

We would like to thank you for your time and effort. Your comments during the first round of revisions were extremely helpful and the quality of the manuscript benefitted greatly from them. 

Reviewer 5 Report

Comments and Suggestions for Authors

The authors have modified the article according to the suggestions and the manuscript has been improved significantly.

However, editing of English language is still needed.

Comments on the Quality of English Language

Editing of English language is still needed.

Round 3

Reviewer 2 Report

Comments and Suggestions for Authors

As mentioned earlier, the overall status of the manuscript is fine. However, it lacks novelty. The authors have mentioned that the novelty lies in the idea of comparing the different bioactive compounds and legumes in terms of content and functionality. Yet, the comparison is very general and not in-depth and sounds like a general summary. As an example, section "6.2. Comparison of Saponins Across Species" - the authors have listed the different types of species, saponin content and bioactivity. If this was a comprehensive review, then the saponin content would have been a range based on evidence from multiple studies, rather than one value. If this was a true comparative study, then the factors affecting this content across different species would have been discussed in-depth (starting from the molecular level all the way to agricultural practices). Moreover, the authors have just listed briefly some bioactive properties of the saponins from different species. In one of the cases, they mentioned "anti-cancer" in the table for soy and mung bean, and in the text, they mentioned "anti-proliferative effects on certain human cancer cells by blocking cell cycle progression". This is very vague, a proper discussion of this topic will include a complete description of the anti-cancer activity specifying the cells and the concentration of saponins to show the effect and the degree of this effect, moreover, this would have then been compared between soy and mung bean. 

The above example is just one of the several cases with respect to the point I mentioned. I can go forever and list these issues. Anyway, I am happy with the manuscript being accepted, however, the authors need to understand that their work is just a summary of the literature and not a comprehensive comparison.